# Effects of IL-1β, IL-20, and BMP-2 on Intervertebral Disc Inflammation under Hypoxia

**DOI:** 10.3390/jcm9010140

**Published:** 2020-01-04

**Authors:** Yu-Hsiang Hsu, Ruey-Mo Lin, Yi-Shu Chiu, Wen-Lung Liu, Kuo-Yuan Huang

**Affiliations:** 1Institute of Clinical Medicine, College of Medicine, National Cheng Kung University, Tainan 701, Taiwan; brianhsu@mail.ncku.edu.tw; 2Clinical Medicine Research Center, National Cheng Kung University Hospital, College of Medicine, National Cheng Kung University, Tainan 701, Taiwan; 3Department of Orthopedics, Tainan Municipal An-Nan Hospital-China Medical University, Tainan 709, Taiwan; d71081@mail.tmanh.org.tw; 4Department of Biochemistry and Molecular Biology, College of Medicine, National Cheng Kung University, Tainan 701, Taiwan; abookchiu@gmail.com; 5Department of Orthopedics, National Cheng Kung University Hospital, College of Medicine, National Cheng Kung University, Tainan 701, Taiwan; manga0522@gmail.com

**Keywords:** inflammatory cytokines-20 (IL-20), intervertebral disc (IVD), inflammation

## Abstract

Intervertebral disc (IVD) is an avascular tissue under hypoxic condition after adulthood. Our previous data showed that inflammatory cytokines (interleukin (IL)-1β), IL-20, and bone morphogenetic protein-2 (BMP-2) play important roles in the healing process after disc injury. In the current study, we investigated whether IL-1β, IL-20, or BMP-2 modulate the expression of pro-inflammatory cytokines, chemotaxis factor, and angiogenesis factor on IVD cells under hypoxia. IVD cells were isolated from patients with intervertebral disc herniation (HIVD) at the levels of L4–5 and L5–S1. We found that the expression of IL-1β, IL-20, BMP-2, hypoxia-inducible factor (HIF)-1α, IL-6, IL-8, angiogenetic factor (vascular endothelial growth factor (VEGF)), chemotactic factor (monocyte chemoattractant protein 1 (MCP-1)), and matrix metalloproteinase-3 (MMP-3) was upregulated in IVD cells under hypoxia conditions. In addition, IL-1β upregulated the expression of pro-inflammatory cytokines (IL-6 and IL-8), VEGF, MCP-1, and disc degradation factor (MMP-3) in IVD cells under hypoxia conditions. IL-20 upregulated MCP-1 and VEGF expression. BMP-2 also upregulated the expression of MCP-1, VEGF, and IL-8 in IVD cells under hypoxia conditions. Treatment with antibody against IL-1β decreased VEGF and MMP-3 expression, while treatment with IL-20 or BMP-2 antibodies decreased MCP-1, VEGF, and MMP-3 expression. Moreover, IL-1β modulated both the expression of IL-20 and BMP-2, but IL-20 only modulated BMP-2 either under a hypoxic or normoxic condition. Therefore, we concluded that the inflammation, chemotaxis, matrix degradation, and angiogenesis after disc herniation are influenced by the hypoxic condition and controlled by IL-1β, IL-20, and BMP-2.

## 1. Introduction

Low back pain and sciatica are the common symptoms in patients with intervertebral disc herniation (HIVD) and degeneration [1,2]. Intervertebral discs (IVDs) degeneration is an important cause of neck, back, and radicular pain [3]. Spinal degeneration originates predominantly from annular injury and nucleus herniation, which is because of the loss of the extracellular matrix constituent proteoglycans, type II collagen, and aggrecans in the nucleus pulposus (NP) [4,5,6,7]. In the pathogenesis of HIVD, the NP has mechanical and chemical effects [6,7]. When IVD is herniated, the protruded NP causes a mechanical compression to the nerve root, and then leads to nerve–root hyperemia, edema, and sciatica. However, mechanical compression is not the only effect involved in HIVD, because HIVD has also been identified in many asymptomatic individuals [8]. The exposed NP is misrecognized as a foreign substance and induces macrophage migration and accumulation into the epidural space, which activates an inflammatory response cascade that results in degradation and resorption of herniated disc material [6,7,9,10,11]. However, the molecular mechanism of the autoimmune reaction induced by the exposed NP after disc herniation is still unclear. Whether several key cytokines trigger inflammatory cascade and promote angiogenesis in the degenerative process of IVD is still unclear and needs to be clarified.

Bone morphogenetic proteins (BMPs) have the potential to induce ectopic bone formation [12]. BMPs are critical in the development of embryo and postnatal growth [13,14]. BMP-2 is particularly crucial in skeletal development, cartilage, and bone formation, and in the process of bone healing [15,16,17,18]. In our previous study, the injured disc degenerated worse in the rhBMP-2-treated group compared with the saline-treated control group, rhBMP-2 plays a critical role in the injury response of the IVD after an annular tear, and we found that angiogenesis and inflammation were prominently induced by BMP-2 after disc injury [19].

Interleukin-20 (IL-20) is involved in psoriasis, atherosclerosis, and rheumatoid arthritis [20,21,22]. IL-20 was expressed in activated monocytes, keratinocytes, synovial fibroblasts, endothelial cells, and several squamous cell carcinomas including the skin, lung, and esophagus [23]. IL-20 actives the signal transducer and activator of transcription 3 (STAT3) pathway through heterodimeric receptor complexes including IL-20R1/IL-20R2 and IL-22R1/IL-20R2 [24]. Our previous study [25] reported that IL-20 and its receptors were expressed in human HIVDs. IL-20 induced interleukin-1β (IL-1β), interleukin-6 (IL-6), monocyte chemoattractant protein 1 (MCP-1), vascular endothelial growth factor (VEGF), and matrix metalloproteinase-3 (MMP-3) expression. IL-1β induced IL-1β, IL-6, IL-8, VEGF, MMP-3, and MCP-1 expression. IL-20 combined with IL-1β synergistically upregulated tumor necrosis factor-α (TNF-α), IL-1β, IL-6, IL-8, MMP-3, and MCP-1 expression. IL-20 acted on inflamed human IVD tissues through autocrine manner to regulate the inflammatory response and angiogenesis in the healing process of herniated discs or the degenerative process of IVDs [25]. 

A previous study found that IL-1β seems to play a more critical role than TNF-α because IL-1β is expressed and produced at higher levels than TNFα in degenerated IVD cells, suggesting IL-1β may be more predominant in the processes of IVD degeneration [26]. IL-1β and its receptor are highly upregulated in IVD degeneration, therefore, it might be a critical mediator in the processes of IVD degeneration [26]. On the basis of the above-mentioned, we hypothesized IL-1β, IL-20, and BMP-2 may play important roles in the healing process after disc injury. IVD is an avascular structure, so we aimed to investigate whether IL-1β, IL-20, or BMP-2 modulates the expressions of proinflammatory cytokines, chemotaxis, and angiogenesis under hypoxia.

## 2. Materials and Methods 

### 2.1. Reagents, Antibody, and Protein Preparation

Human recombinant IL-20 protein, anti-IL-20 monoclonal antibody, and isotype IgG1 antibody were purchased from R&D Systems (Minneapolis, MN, USA), and anti-IL-1β and anti-bone morphogenetic protein-2 (BMP-2) monoclonal antibodies were purchased from Sigma (St. Louis, MO USA). Recombinant human IL-1β and BMP-2 were purchased from PeproTech (Rocky Hill, NJ, USA).

### 2.2. Immunohistochemical (IHC) Staining

Paraffin-embedded sections were deparaffinized, rehydrated, and antigen-retrieval with Tris- ethylenediaminetetraacetic acid (EDTA) buffer for immunohistochemistry staining. Tissue sections were then incubated in antibody diluent (#S3022, DAKO, Carpenteria, CA, USA) and stained with anti-IL-20, anti-IL-1β, and anti-BMP-2 monoclonal antibodies at 4 °C overnight. Incubating tissue sections with mouse IgG isotype instead of primary antibody were the negative control. The immune reactivity of positive staining was developed using the 3-amino-9-ethylcarbazole (AEC) chromogen kit (Romulin AEC Chromogen Kit, Biocare Medical, Walnut Creek, CA, USA) and counterstained with Mayer’s hematoxylin.

### 2.3. Primary Culture of Human IVD Cells

IVD cells were isolated from 10 patients with HIVD at the levels of L4–5 and L5–S1. The primary culture of IVD cells from herniated disc was mentioned as previously described [26,27]. The disc tissues were digested with 0.2% collagenase type II (Worthington, Lakewood, CO, USA) in Dulbecco’s modified eagle medium/nutrient mixture F-12 (DMEM/F12) medium (Gibco, Grand Island, NY, USA) for 6 h. The isolated cells were cultured in DMEM/F12 medium supplemented with 10% fetal bovine serum (FBS), 100 U/mL penicillin, 100 µg/mL streptomycin, and 2 mM L-glutamine (Gibco, Grand Island, NY, USA). Cells were used for experiments between three and eight passages. The National Cheng Kung University Hospital Institutional Review Board approved the study (IRB number: ER-95-136). Signed informed consent was obtained from all participants. 

### 2.4. Experimental Culture Conditions for Hypoxia Stimulation

The experiments were performed using a standard Plexiglas chamber (Bellco Glass, Vineland, NJ, USA) deoxygenated by positive infusion of a 5% carbon dioxide/95% nitrogen gas mixture. Cells were starved for 0.5 h in serum-free DMEM/F12 medium and then placed in the hypoxia chamber. Monitoring infusion with a standardized pressure gauge ensured equal atmospheric pressure. During the experiment, cell cultures were placed in a standard humidified tissue incubator at 37 °C, with oxygen saturation kept below 1%, and they were continuously monitored. Total cellular RNA was then isolated after the indicated time course.

### 2.5. Reverse Transcription and Real-Time Quantitative Polymerase Chain Reaction (qPCR)

IVD cells (1 × 10^6^ cells/well) were cultured in serum-free DMEM/F12 medium under normoxia and hypoxia conditions, treated with or without IL-1β (10 ng/mL), IL-20 (200 ng/mL), and rhBMP-2 (200 ng/mL), and were harvested at different time points. For neutralization antibody experiment, anti-IL-1β (2 μg/mL), anti-IL-20 (2 μg/mL), anti-BMP-2 (2 μg/mL) antibodies, or control mIgG (2 μg/mL) (R&D Systems, #20102) were treated with cells for 6 h. Total RNA was extracted using Trizol reagent (Life Technologies, Carlsbad, CA, USA) and then underwent reverse transcription according to the manufacturer’s instructions. The mRNA expression of IL-1β, IL-6, IL-8, BMP-2, type II receptor for bone morphogenetic protein (BMPRII), VEGF, hypoxia-inducible factor 1-α (HIF-1α), MMP-3, melanocyte-inhibiting factor-1 (MIF-1), and MCP-1 under hypoxic condition was analyzed with gene-specific primers. Detection of amplified templates was performed using Applied Biosystems StepOnePlus detection system with SYBR Green I (Thermo Fisher Scientific, Waltham, MA, USA). To calculate the relative expression level of the target genes, we normalized to the gene encoding β-actin and compared it with the mean values of control.

### 2.6. Enzyme-Linked Immunosorbent Assay (ELISA) Analysis

Primary disc cells (1 × 10^6^ cells/well) were cultured in serum-free DMEM/F12 and treated under normoxia or hypoxia with phosphate-buffered saline (PBS), control IgG (2 μg/mL), anti-IL-20 (2 μg/mL), anti-IL-1β (2 μg/mL), or anti-BMP-2 (2 μg/mL) for 24 h. The levels of VEGF or MMP-3 in the conditioned medium were measured using ELISA kits according to the manufacturer’s instructions (R&D Systems, Minneapolis, MN, USA).

### 2.7. Statistical Analysis

Prism 8.0 (GraphPad Software, La Jolla, California, USA) was used for the statistical analysis. A one-way analysis of variance (ANOVA) nonparametric test (Kruskal–Wallis test) was used to compare the data between groups. Data are expressed as the mean of replicate measurements or mean normalized values between multiple experiments ± SEM or SD. *p* < 0.05 was considered statistically significantly.

## 3. Results

### 3.1. Hypoxia Effect On Primary Cultured Disc Cells

IHC staining confirmed that IL-20, IL-1β, and BMP-2 were positively stained in intervertebral disc (IVD) sections from patients with HIVD (Figure 1A). RT-qPCR showed that the hypoxia-inducible factor-1 (HIF-1α), BMP-2, pro-inflammatory cytokines (IL-1β, IL-6, IL-8, and IL-20), chemokine (MCP-1), angiogenesis-associated gene VEGF, and disc degradation-associated factor MMP-3 were upregulated in primary cultured IVD cells (Figure 1B–C). The mRNA expression of IL-1β and IL-20 was upregulated rapidly and constituted in response to hypoxia conditions. RT-qPCR also showed that IL-20’s receptors IL-20R1 and IL-20R2 were upregulated in primary cultured IVD cells under hypoxia conditions (Figure 1D). There was no statistically significant difference of the expression of BMP-2’s receptor, BMPRII, between normoxia and hypoxia conditions (data not known). These data indicated that several critical factors were upregulated in primary cultured IVD cells under hypoxic conditions. 

### 3.2. The Effects of IL-20, IL-1β, and BMP-2 in Primary Cultured Disc Cells under Hypoxia

To investigate whether IL-1β, IL-20, or BMP-2 modulates the expression of pro-inflammatory cytokines, chemotactic factor, and angiogenetic factor in IVD cells under hypoxia, we performed RT-qPCR and showed that IL-1β upregulated the expression of pro-inflammatory cytokines (IL-6 and IL-8), angiogenetic factor (VEGF), chemotactic factor (MCP-1), and disc degradation factor (MMP-3) in IVD cells under hypoxia conditions. IL-20 upregulated MCP-1 and VEGF expression. BMP-2 upregulated the expression of MCP-1, VEGF, and IL-8 in IVD cells under hypoxia conditions (Figure 2A–F).

### 3.3. Treatment with Antibodies Against IL-1β, IL-20, and BMP-2 on Disc Cells under Hypoxia

On the basis of our observation mentioned above, IL-1β, IL-20, and BMP-2 regulated these gene expressions under hypoxia; we hypothesized that IL-1β, IL-20, and BMP-2 might be the upstream mediators in response to hypoxia. Therefore, specific antibodies against IL-1β, IL-20, and BMP-2 might be the strategy to reverse the hypoxia-induced effects in IVD cells. Antibodies against IL-1β, IL-20, and BMP-2 had no effect on the HIF-1α expression in IVD cells under hypoxia (Figure 3A). However, RT-qPCR showed that treatment with antibody against IL-1β decreased VEGF and MMP-3 expression, while treatment with IL-20 or BMP-2 antibodies decreased MCP-1, VEGF, and MMP-3 expression (Figure 3B–D). These data suggested that the upregulation of MMP-3 and VEGF was indeed mediated by IL-20, IL-1β, and BMP-2.

For further confirmation of the RT-qPCR results, the protein levels of VEGF and MMP-3 in the conditioned culture medium of the human IVD cells were determined. We found that IL-1β antibody has slightly reduced the protein expression of VEGF, however, there is no significant difference. IVD cells incubated with IL-20 antibody significantly decreased the protein level of VEGF. Nevertheless, VEGF levels declined dramatically in response to BMP-2 antibody treatment (Figure 4A). In addition, the protein level of MMP-13 was inhibited in IL-20, IL-1β, or BMP-2 antibodies treatment (Figure 4B). 

### 3.4. The Relationship Between IL-1β, IL-20, and BMP-2

To further clarify the regulation between IL-1β, IL-20, and BMP-2 in IVD cells, we incubated IVD cells with IL-1β, IL-20, or BMP-2 under hypoxia or normoxia. RT-qPCR showed that IL-1β induced the expression of IL-20 and BMP-2 in IVD cells under hypoxic and normoxic conditions (Figure 5A–B). In addition, IL-20 induced the expression of BMP-2, but not IL-1β in IVD cells under hypoxic and normoxic conditions (Figure 5C–D). BMP-2 did not induce IL-1β or IL-20 expression in IVD cells under hypoxic and normoxic conditions (Figure 5E–F).

## 4. Discussion

In the present study, we found many proinflammatory cytokines were upregulated in IVD cells under hypoxia conditions. IL-1β upregulated the expression of pro-inflammatory cytokines (IL-6 and IL-8), angiogenetic factor (VEGF), chemotactic factor (MCP-1), and disc degradation factor (MMP-3) in IVD cells under hypoxia conditions. IL-20 upregulated MCP-1 and VEGF expression. BMP-2 also upregulated the expression of MCP-1, VEGF, and IL-8 in IVD cells under hypoxia conditions. In addition, IL-1β modulated both the expression of IL-20 and BMP-2, but IL-20 only modulated BMP-2 either under a hypoxic or normoxic condition. Therefore, we concluded that the inflammation, chemotaxis, matrix degradation, and angiogenesis after disc herniation are influenced by the hypoxic condition and controlled by IL-1β, IL-20, and BMP-2.

In the current study, we are interested in the effect of IL-1β, IL-20, and BMP-2 on IVD cells under hypoxia. Although the expression of HIF-1α in disc cells was induced under hypoxic condition, we also found the expression of HIF-1α under normoxic condition. This phenomenon had also been noted by A. Agrawal and his colleagues (2007) [28]. They found that HIF-1α is a transcription factor that transactivates the expression of target genes in regulating energy metabolism (glycolysis), aggrecan synthesis, and vascularization [28,29]. E. Schipani, et al. also demonstrated that HIF-1α acted as a survival factor in chondrocytes of growth plate, another avascular tissue like nucleus pulposus in intervertebral disc, which increases the expression of VEGF and enzymes’ activity of the glycolytic metabolism under the hypoxic condition. While the cells lack HIF-1α (deletion with tissue-specific targeting), the growth arrest of chondrocyte altered (owing to the decreased expression of cyclin-dependent kinase inhibitor p57 by HIF-1α deletion) and cause cell death under the hypoxic condition [30]. Therefore, it could be hypothesized that HIF-1α in the nucleus pulposus cells of intervertebral disc could have acted in the same manner as in the chondrocyte of growth plate to maintain the cell survival under hypoxia. Meng, X.C., et al. found that knockout of HIF-1α leads to IVD degeneration in mice [31].

For the pathogenesis of herniated disc, A. Minamide, et al, (1999) proposed that NP, the central composition of IVD, plays an important role as biochemical factors in addition to its role in mechanical compression [32]. Nerve root pain may be caused by vascular and inflammatory infiltration. Kato, T, et al. (2004) indicated that the increased amount of TNFα would stimulate the secretion of VEGF, and thus induce angiogenesis. Angiogenesis is critical in the degradation of the herniated nucleus [33]. In this study, we focused on IVD inflammation, chemotaxis, and angiogenesis under a hypoxic condition. Furthermore, the roles of IL-20 and BMP-2 under these conditions are investigated. We previously reported that angiogenesis and inflammation were prominently induced by BMP-2 after disc injury [19]. Nevertheless, whether BMP-2 and its receptor were expressed in human IVD and articular cartilage has not yet been reported. In our present study, we found that the primarily cultured human disc cells could be the target cells for BMP-2 in HIVD tissue. Macrophages and other inflammatory cells infiltrated into herniated disc tissues have been reported and, in fact, IL-8 and MCP-1 are important chemokines for immune cell chemotaxis [9]. IL-8 promotes chemotaxis of neutrophils and angiogenesis [34]. Therefore, BMP-2 might have contributed to angiogenesis after disc herniation, via the expression of IL-8 and VEGF. In the present study, we found that BMP-2 affected VEGF expression in primary IVD cells. Therefore, BMP-2 might have two effects including promoting inflammation and regulating angiogenesis in herniated disc tissues. 

A previous study showed that IL-20 promoted angiogenesis directly or indirectly through VEGF [35]. In the present study, we also found that IL-20 affected VEGF expression in primary IVD cells. Therefore, IL-20 might also be involved in inflammation and in the regulation of angiogenesis in herniated disc tissues. According to our data and some articles we have reviewed recently, we might be able provide some explanation about the presentation of IL-20 in HIVD. Under the normal circumstances, the NP was surrounded by annulus fibrosus in a hypoxic environment, HIF-1α was expressed to sustain its survival, but when the disc is herniated, protruded NP that exposed under hypoxic environment would express MCP-1 as chemotaxis to recruit the macrophage there. K. Murai and his colleagues have directly demonstrated that the macrophage and natural killer (NK) cells would have a cytotoxic effect on protruded NP cells through an in vitro cytotoxicity assay and in vivo study [36]. Their findings suggested that the macrophage and NK cells were recruited to eliminate and reabsorb the protruded NP in the acute inflammatory response, and no T cells were involved in it. 

Although hypoxia conditions upregulated many factors in IVD cells, anti-IL-1β, anti-IL-20, and anti-BMP-2 antibodies significantly inhibited the MMP-3 and VEGF protein levels in IVD cells under hypoxia, which indicated that the regulation of MMP-3 and VEGF was indeed mediated by IL-1β, IL-20, and BMP-2. In addition, we found that IL-1β induced the expression of IL-20 and BMP-2, and IL-20 induced BMP-2, but not IL-1β. BMP-2 did not induce IL-1β and IL-20. These data indicated that IL-1β might be the upstream of IL-20 and BMP-2. 

There were some limitations in this study. First, the intervertebral disc theoretically bears the axial force under gravity in the actual situation, so the IVD cells should be compressed by the axial load to mimic the loading of gravity; therefore, a future study is needed to investigate the cyclic compressive or tensile stress on the IVD cells. Second, we investigated the effect of IL-1β, IL-20, and BMP-2 on the expression of pro-inflammatory cytokines, chemotaxis factor, and angiogenesis factor of IVD cells under hypoxia. Our data are an accumulation of phenomenology; however, the weakness of this research was that we did not investigate the signaling pathway or possible molecular mechanism to address the roles of these cytokines in the pathogenesis of HIVD, which awaits future investigation.

## 5. Conclusions

Our data demonstrate that IL-1β, IL-20, and BMP-2 in combination with hypoxia could induce cytokines or chemokines expression and regulate angiogenetic factors in the pathogenesis of disc herniation. Therefore, the inflammation, chemotaxis, matrix degradation, and angiogenesis after disc herniation are influenced by the hypoxic condition and controlled by IL-1β, IL-20, and BMP-2. Targeting IL-1β, IL-20, and BMP-2 may be a novel strategy for alleviating inflammation and angiogenesis in patients with HIVD. 

## Figures and Tables

**Figure 1 jcm-09-00140-f001:**
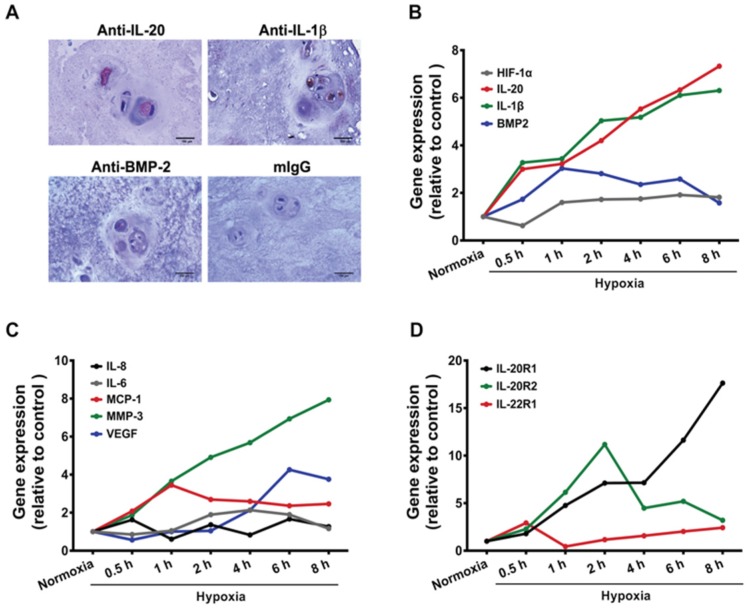
Effects of hypoxia stimulation on gene expression of intervertebral disc (IVD) cells. (**A**) Expression of interleukin (IL)-20, IL-1β, and bone morphogenetic protein (BMP)-2 in intervertebral disc herniation (HIVD) sections were determined using immunohistochemical (IHC) staining with specific antibodies. Staining with isotype mouse immunoglobulin G (IgG) was used as the negative control. Scale bars = 100 μm. All experiments were performed three times with similar results. Data are from a representative experiment. (**B**–**D**) Primary cultured IVD cells (1 × 10^6^ cells/well) were exposed to a hypoxic environment for 0.5, 1, 2, 4, 6, and 8 h (h). The expression levels of indicated genes were analyzed using real-time quantitative polymerase chain reaction (RT-qPCR) with specific primers. β-actin was an internal control. Data are expressed as mean and are representative of three independent experiments. VEGF, vascular endothelial growth factor; MMP-3, matrix metalloproteinase-3; MCP-1, monocyte chemoattractant protein 1.

**Figure 2 jcm-09-00140-f002:**
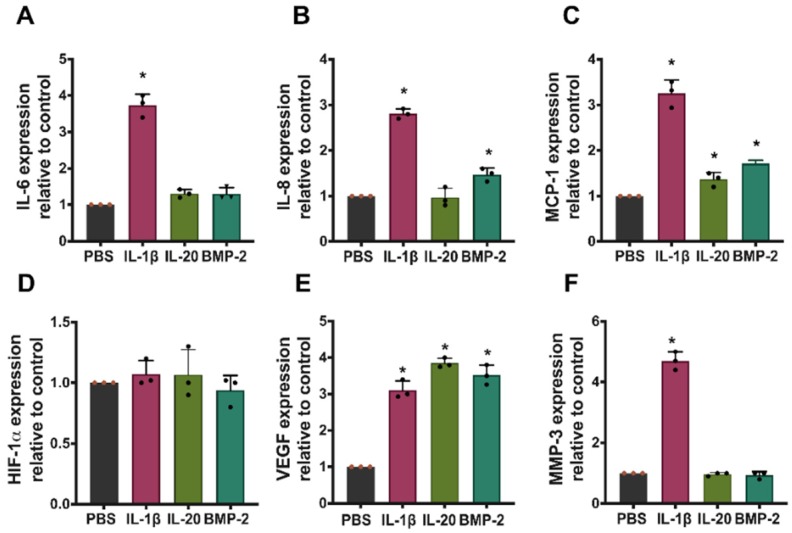
Effects of IL-1β, IL-20, and BMP-2 in IVD cells under hypoxia. IVD cells were exposed to phosphate-buffered saline (PBS), IL-1β (10 ng/mL), IL-20 (200 ng/mL), and BMP-2 (200 ng/mL) under hypoxia for six hours. (**A**–**F**) The expression levels of indicated genes were analyzed using RT-qPCR with specific primers. β-actin was an internal control. Dots on the bar chart indicated the triplicates for each group. * *p* < 0.05 compared with PBS-treated controls. Data are expressed as mean ± SD and are representative of three independent experiments.

**Figure 3 jcm-09-00140-f003:**
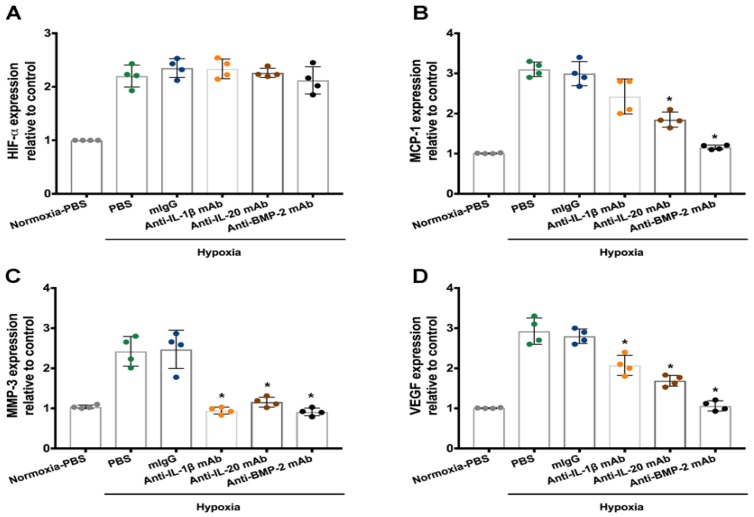
The effects of anti-IL-1β, -IL-20, and -BMP-2 mAb in IVD cells under hypoxia. IVD cells were exposed to PBS, mIgG (2 μg/mL), IL-1β antibody (2 μg/mL), IL-20 antibody (2 μg/mL), and BMP-2 antibody (2 μg/mL) under hypoxic conditions for six hours (*n* = 4/group). (**A**–**D**) The expression levels of indicated genes were analyzed using RT-qPCR with specific primers. β-actin was an internal control. Dots on the bar chart indicated the quadruplicates for each group. * *p* < 0.05 compared with mIgG-treated controls. Data are expressed as mean ± SD and are representative of three independent experiments.

**Figure 4 jcm-09-00140-f004:**
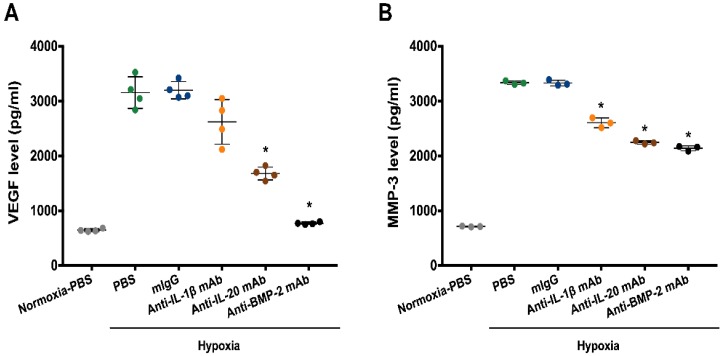
MMP-3 and VEGF levels in anti-IL-1β, -IL-20, and -BMP-2 mAb-treated IVD cells. IVD cells treated with antibodies against IL-1β, IL-20, and BMP-2 under hypoxia. Primary cultured IVD cells were exposed to PBS, mIgG (2 μg/mL), IL-1β antibody (2 μg/mL), IL-20 antibody (2 μg/mL), and BMP-2 antibody (2 μg/mL) under hypoxic conditions for 24 h (*n* = 4/group). (**A**,**B**) The conditioned medium was then collected and analyzed using MMP-3 and VEGF enzyme-linked immunosorbent assay (ELISA) kits. Dots on the bar chart indicated the triplicates or quadruplicates for each group. * *p* < 0.05 compared with mIgG-treated controls. Data are expressed as mean ± SD. The experiment was repeated twice with similar results.

**Figure 5 jcm-09-00140-f005:**
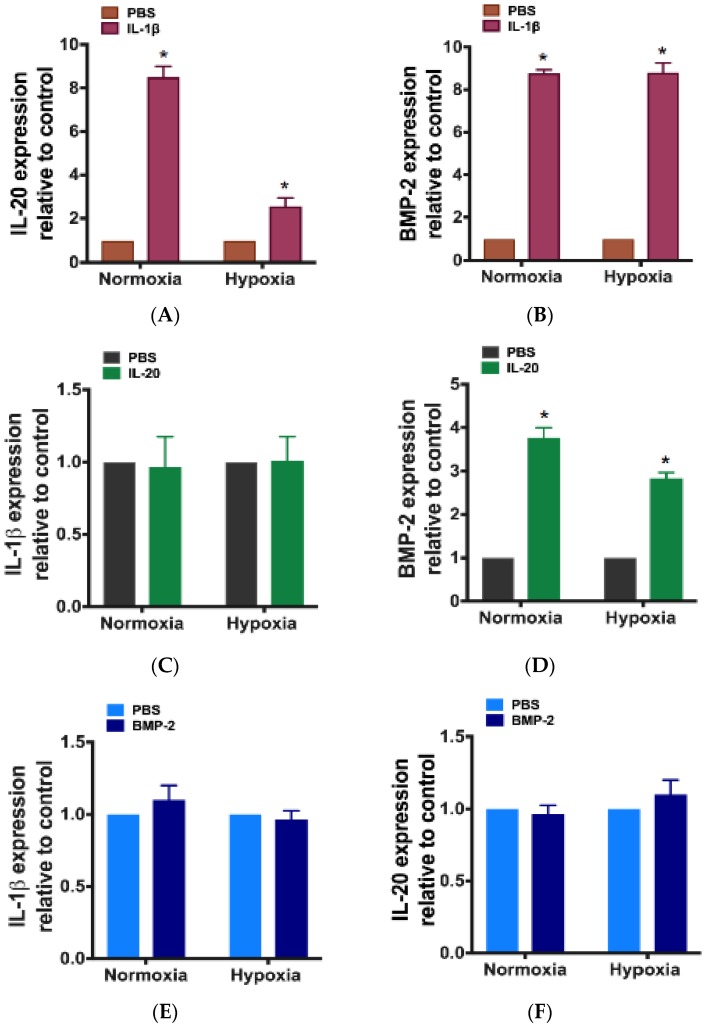
The regulation of IL-1β, IL-20, and BMP-2 in IVD cells. IVD cells were exposed to PBS, IL-1β (10 ng/mL), IL-20 (200 ng/mL), and BMP-2 (200 ng/mL) under normoxia or hypoxia for six hours. (**A**–**F**) The expression levels of indicated genes were analyzed using RT-qPCR with specific primers. β-actin was an internal control. * *p* < 0.05 compared with PBS-treated controls. Data are expressed as mean ± SD and are representative of three independent experiments.

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
