# Peer review of "Effects of IL-1β, IL-20, and BMP-2 on Intervertebral Disc Inflammation under Hypoxia"

_jcm, 2020, doi:10.3390/jcm9010140_

Round 1

Reviewer 1 Report

Major comments

This research investigated the effect of IL1b/IL20/BMP-2 on intervertebral disc inflammation cytokines under hypoxia. The strength of this research was the target was the primary human IVD cells. Your data was excellent and statistical analysis was almost perfect. But your data is an accumulation of phenomenology and you did not investigate the mechanism.

We understand that IL1b-IL20-BMP-2 axis was important for VEGF expression, for example. However, “In the present study, we found that BMP-2 induced VEGF expression in disc cells.” is scientifically incorrect. You only showed the association between the presence of BMP-2 and VEGF expression.

I agree with the second to fourth experiments were performed under hypoxia for 6 hours. But there is no data of 6 hours in the Figure 1, although there is a data about 6 hours incubation.

Minor comments

Figure 1.

A: The name of antibody should be bigger.

B-D: The word “hypoxia” was used repeatedly. You should change this expression on X-axis concisely.

Figure 2: The concentration of cytokines was united, therefore you also should change this expression concisely and explain the concentration in the figure legend.

Is this one experience? If not, you should use plot figure like Fig3/4.

Figure 3 and 4: You should insert the sentence “n= ? biologically independent experiments” in the figure legends.

Figure 5. This figure is so called “gain of function.” The data about VEGF is needed. If so, the discussion would be better.

Author Response

Reviewer #1

This research investigated the effect of IL1b/IL20/BMP-2 on intervertebral disc inflammation cytokines under hypoxia. The strength of this research was the target was the primary human IVD cells. Your data was excellent and statistical analysis was almost perfect. But your data is an accumulation of phenomenology and you did not investigate the mechanism.

REPLY:

We agree with you that our data is an accumulation of phenomenology and it is an important issue to further investigate the mechanism of these cytokines in the pathogenesis of HIVD. However, it is difficult to collect and isolate enough primary mature human IVD cells to investigate the possible mechanism in this study. We will finish this part in our next manuscript in the near future. As suggested, we have stated the limitations of this study in our revised manuscript, as follows: (Lines, 280-288)

“There were some limitations in this study: First, the intervertebral disc theoretically bears the axial force under gravity in the actual situation, so the IVD cells should be compressed by the axial load to mimic the loading of gravity, therefore, a future study is needed to investigate the cyclic compressive or tensile stress on the IVD cells. Second, we investigated the effect of IL-1β, IL-20, and BMP-2 on the expression of pro-inflammatory cytokines, chemotaxis factor, and angiogenesis factor of IVD cells under hypoxia. Our data is an accumulation of phenomenology; however, the weakness of this research was that we did not investigate the signaling pathway or possible molecular mechanism to address the roles of these cytokines in the pathogenesis of HIVD, which awaits future investigation.”

We understand that IL1b-IL20-BMP-2 axis was important for VEGF expression, for example. However, “In the present study, we found that BMP-2 induced VEGF expression in disc cells.” is scientifically incorrect. You only showed the association between the presence of BMP-2 and VEGF expression.

REPLY:

Thanks for your suggestion, we agreed with your valuable suggestion that “ In the present study, we found that BMP-2 induced VEGF expression in disc cells.” is scientifically incorrect. Because we only showed the association between the presence of BMP-2 and VEGF expression.” And we have reorganized and edited the sentences and more conservatively discussed our findings in the revised manuscript as follows:

Section of Discussion (Lines, 257-259):

“Therefore, BMP-2 might have contributed to angiogenesis after disc herniation, via the expression of IL-8 and VEGF. In the present study, we found that BMP-2 affected VEGF expression in primary IVD cells.”

I agree with the second to fourth experiments were performed under hypoxia for 6 hours. But there is no data of 6 hours in the Figure 1, although there is a data about 6 hours incubation.

REPLY:

As suggested, we have added these data and redrawn the Figure 1 in the revised manuscript in the section of results as follows (Lines, 150-159):

Figure 1. Effects of hypoxia stimulation on gene expression of IVD cells. (A) Expression of IL-20, IL-1β, and BMP-2 in HIVD sections were determined using IHC staining with specific antibodies. Staining with isotype mouse IgG was used as the negative control. Scale bars = 100 μm. All experiments were performed three times with similar results. Data are from a representative experiment. (B-D) Primary cultured IVD cells (1×106 cells/well) were exposed to a hypoxic environment for 0.5, 1, 2, 4, 6, 8 hours (h). Total RNA was isolated for RT-qPCR with specific primers. β-actin was an internal control. Data are expressed as mean and are representative of three independent experiments.

Figure 1. A: The name of antibody should be bigger.

REPLY:

As suggested, we have redrawn the figure.

Figure 1. B-D: The word “hypoxia” was used repeatedly. You should change this expression on X-axis concisely.

REPLY:

 As suggested, we have redrawn the figure.

Figure 2: The concentration of cytokines was united, therefore you also should change this expression concisely and explain the concentration in the figure legend.

Is this one experience? If not, you should use plot figure like Fig3/4.

REPLY:

 We performed this experiment for three times. We described in the figure 1 as “All experiments were performed three times with similar results.”

As suggested, we have redrawn the figure 2. (Lines, 167-173)

Figure 2. Effects of IL-1β, IL-20, and BMP-2 in IVD cells under hypoxia. IVD cells were exposed to PBS, IL-1β (10 ng/ml), IL-20 (200 ng/ml), and BMP-2 (200 ng/ml) under hypoxia for 6 hours. (A-F) Total RNA was then isolated for RT-PCR with specific primers. β-actin was an internal control. *P < 0.05 compared to PBS-treated controls. Data are expressed as mean ± SD and are representative of three independent experiments.

Figure 3 and 4: You should insert the sentence “n= ? biologically independent experiments” in the figure legends.

REPLY:

As suggested, we have added this information (n = 4 /group) and briefly described in the revised manuscript as follows: (Lines, 183-190 and Lines, 197-205)

Figure 3. The effects of anti-IL-1β, -IL-20 and -BMP-2 mAb in IVD cells under hypoxia. IVD cells were exposed to PBS, mIgG (2 μg/ml), IL-1β antibody (2 μg/ml), IL-20 antibody (2 μg/ml) and BMP-2 antibody (2 μg/ml) under hypoxic conditions for 6 hrs (n = 4 /group). (A-D) Total RNA was isolated for RT-qPCR with specific primers. β-actin was an internal control. *P < 0.05 compared to mIgG-treated controls. Data are expressed as mean ± SD and are representative of three independent experiments.

Figure 4. MMP-3 and VEGF levels in anti-IL-1β, -IL-20 and -BMP-2 mAb-treated IVD cells.  IVD cells treated with antibodies against IL-1β, IL-20 and BMP-2 under hypoxia. Primary cultured IVD cells were exposed to PBS, mIgG (2 μg/ml), IL-1β antibody (2 μg/ml), IL-20 antibody (2 μg/ml) and BMP-2 antibody (2 μg/ml) under hypoxic conditions for 24hrs (n = 4 /group). (A-B) The conditioned medium was then collected and analyzed using MMP-3 and VEGF ELISA kits. *P < 0.05 compared to mIgG-treated controls. Data are expressed as mean ± SD. The experiment was repeated twice with similar results.

Figure 5. This figure is so called “gain of function.” The data about VEGF is needed. If so, the discussion would be better.

REPLY:

Based on our data from Figure 1, IL-20, IL-1B, and BMP-2, but not VEGF were quickly upregulated under hypoxia conditions (0.5 h). Therefore, we speculated that IL-20, IL-1β, and BMP-2 might be the upstream signal of VEGF. Therefore, we only focus and clarify the regulation between IL-20, IL-1β, and BMP-2 in the primary IVD cells.

Reviewer 2 Report

This is an interesting study that I believe is worthy of publication. My queries and suggestions to help improve the manuscript follow:

The sentence starting with 'Bone morphogenetic..." at line 54 appears to be incomplete Some acronyms are not stated in full the first time they are presented (e.g. IL-20 , PBS) The introduction could have also mentioned what the gaps in the literature are to help justify why the study was completed At line 101 it would be interesting to know how many patients donated cells for investigation and if possible provide some demographic information Lines 108 and 109 repeat the same thing The sentence starting with "to investigate..." on line 148 appears to be incomplete. In the results section information is presented that appears to be more suited to the methods section (e.g. line 149 ... First, we incubated....) or the discussion (e.g. line 154 Previous study indicated...). There are a number of instances throughout where I think the authors should review the information in each section to ensure it is appropriate for that particular section. There are some simple typos throughout that should be corrected (e.g. line 154 'at last for 8 hours', line 157...was no change...) The figure captions appear to be very long and include information that is suited to the methods section. I would suggest reviewing all figure captions The sentence starting at line 170 appears to be incomplete and is again followed by information that should appear in the methods. Sentence starting in line 184 is incomplete I'm not 100% sure what the dots represent near the tops of the bars in the figures. I would like to see the discussion section focusing on the discussion of the main findings. It begins by describing related concepts instead of the findings and what they may mean. I'd like to see the authors adding a section that acknowledges the limitations of the study There are no references of material after 2014 which suggests that the material discussed may not be current. Are there no more recent studies that are relevant to this topic area?

Author Response

Reviewer #2:

The sentence starting with 'Bone morphogenetic..." at line 54 appears to be incomplete Some acronyms are not stated in full the first time they are presented (e.g. IL-20 , PBS)

REPLY:

Thanks for your suggestion, some acronyms which previously were not stated as full statement in the first time have been corrected to the full statement as follows:

 “Bone morphogenetic proteins (BMPs) have the potential to induce ectopic bone formation [12]. “

 “Interleukin-20 (IL-20) is involved in psoriasis, atherosclerosis, and rheumatoid arthritis [20-22]. “

“phosphate-buffered saline (PBS),”

The introduction could have also mentioned what the gaps in the literature are to help justify why the study was completed

REPLY:

Thanks for your suggestion, we have added the sentences “However, the molecular mechanism of the autoimmune reaction induced by the exposed NP after disc herniation is still unclear. Whether several key cytokines trigger inflammatory cascade and promote angiogenesis in the degenerative process of IVD is still unclear and need to be clarified.” in the Introduction section in the revised manuscript.

At line 101 it would be interesting to know how many patients donated cells for investigation and if possible provide some demographic information

REPLY:

Thanks for your suggestion, we have added the detail information in Materials and Methods section in the revised manuscript as follows: (Line 96)

IVD cells were isolated from 10 patients with HIVD at the levels of L4-5 and L5-S1,

Lines 108 and 109 repeat the same thing

REPLY:

Thanks for your suggestion, and we have deleted and edited the sentences in the revised manuscript as follows: (Lines, 102-104)

“The National Cheng Kung University Hospital Institutional Review Board approved the study (IRB number: ER-95-136). Signed informed consent was obtained from all participants.”

The sentence starting with "to investigate..." on line 148 appears to be incomplete. REPLY:

Thanks for your suggestion, and we have edited the sentences in the revised manuscript as follows: (Lines, 140-144)

“IHC staining confirmed that IL-20, IL-1β, and BMP-2 were positively stained in intervertebral disc (IVD) sections from patients with HIVD (Figure 1A). RT-qPCR showed that the hypoxia-inducible factor-1 (HIF-1α), BMP-2, pro-inflammatory cytokines (IL-1β, IL-6, IL-8, and IL-20), chemokine (MCP-1), angiogenesis-associated gene VEGF, and disc degradation-associated factor MMP-3 were upregulated in primary cultured IVD cells (Figure 1B-C). - - -”.

In the results section information is presented that appears to be more suited to the methods section (e.g. line 149 ... First, we incubated....) or the discussion (e.g. line 154 Previous study indicated...). There are a number of instances throughout where I think the authors should review the information in each section to ensure it is appropriate for that particular section.

REPLY:

Thanks for your suggestion, we have reorganized and edited the sentences in the revised manuscript as follows: (Lines, 139-150)

3.1. Hypoxia effect on primary cultured disc cells

IHC staining confirmed that IL-20, IL-1β, and BMP-2 were positively stained in intervertebral disc (IVD) sections from patients with HIVD (Figure 1A). RT-qPCR showed that the hypoxia-inducible factor-1 (HIF-1α), BMP-2, pro-inflammatory cytokines (IL-1β, IL-6, IL-8, and IL-20), chemokine (MCP-1), angiogenesis-associated gene VEGF, and disc degradation-associated factor MMP-3 were upregulated in primary cultured IVD cells (Figure 1B-C). The mRNA expression of IL-1β and IL-20 was upregulated rapidly and constitute in response to hypoxia conditions. RT-qPCR also showed that IL-20’s receptors IL-20R1 and IL-20R2 were upregulated in primary cultured IVD cells under hypoxia conditions (Figure 1D). There was no statistically significant difference of the expression of BMP-2’s receptor, BMPRII between normoxia and hypoxia conditions (data not known). These data indicated that several critical factors were upregulated in primary cultured IVD cells under hypoxic conditions. ”

There are some simple typos throughout that should be corrected (e.g. line 154 'at last for 8 hours', line 157...was no change...)

REPLY:

Thanks for your suggestion, we have corrected some typos in the revised manuscript as follows: (Lines, 139-150)

3.1. Hypoxia effect on primary cultured disc cells

IHC staining confirmed that IL-20, IL-1β, and BMP-2 were positively stained in intervertebral disc (IVD) sections from patients with HIVD (Figure 1A). RT-qPCR showed that the hypoxia-inducible factor-1 (HIF-1α), BMP-2, pro-inflammatory cytokines (IL-1β, IL-6, IL-8, and IL-20), chemokine (MCP-1), angiogenesis-associated gene VEGF, and disc degradation-associated factor MMP-3 were upregulated in primary cultured IVD cells (Figure 1B-C). The mRNA expression of IL-1β and IL-20 was upregulated rapidly and constitute in response to hypoxia conditions. RT-qPCR also showed that IL-20’s receptors IL-20R1 and IL-20R2 were upregulated in primary cultured IVD cells under hypoxia conditions (Figure 1D). There was no statistically significant difference of the expression of BMP-2’s receptor, BMPRII between normoxia and hypoxia conditions (data not known). These data indicated that several critical factors were upregulated in primary cultured IVD cells under hypoxic conditions.   ”

The figure captions appear to be very long and include information that is suited to the methods section. I would suggest reviewing all figure captions

REPLY:

As suggested, we have corrected the figure captions in the revised manuscript as follows: (Lines, 183-190)

Figure 3. The effects of anti-IL-1β, -IL-20 and -BMP-2 mAb in IVD cells under hypoxia. IVD cells were exposed to PBS, mIgG (2 μg/ml), IL-1β antibody (2 μg/ml), IL-20 antibody (2 μg/ml) and BMP-2 antibody (2 μg/ml) under hypoxic conditions for 6 hrs (n = 4 /group). (A-D) Total RNA was isolated for RT-qPCR with specific primers. β-actin was an internal control. *P < 0.05 compared to mIgG-treated controls. Data are expressed as mean ± SD and are representative of three independent experiments.

The sentence starting at line 170 appears to be incomplete and is again followed by information that should appear in the methods. Sentence starting in line 184 is incomplete

REPLY:

Thanks for your suggestion, and we have corrected the sentences in the revised manuscript as follows:  (Lines, 160-167)

3.2. The effects of IL-20, IL-1β, and BMP-2 in primary cultured disc cells under hypoxia

To investigate whether IL-1β, IL-20, or BMP-2 modulates the expression of pro-inflammatory cytokines, chemotactic factor, and angiogenetic factor in IVD cells under hypoxia, we performed RT-qPCR and showed that IL-1β upregulated the expression of pro-inflammatory cytokines (IL-6 and IL-8), angiogenetic factor (VEGF), chemotactic factor (MCP-1), and disc degradation factor (MMP-3) in IVD cells under hypoxia conditions. IL-20 upregulated MCP-1 and VEGF expression. BMP-2 upregulated the expression of MCP-1, VEGF, and IL-8 in IVD cells under hypoxia conditions (Figure 2A-F).”

“3.3. Treatment with antibodies against IL-1β, IL-20 and BMP-2 on disc cells under hypoxia

Based on our observation mentioned above, IL-1β, IL-20, and BMP-2 regulated these gene expressions under hypoxia, we hypothesized that IL-1β, IL-20 and BMP-2 might be the upstream mediators in response to hypoxia. Therefore, specific antibodies against IL-1β, IL-20, and BMP-2 might be the strategy to reverse the hypoxia-induced effects in IVD cells. Antibodies against IL-1β, IL-20, and BMP-2 had no effect on the HIF-1α expression in IVD cells under hypoxia (Figure 3A). However, RT-qPCR showed that treatment with antibody against IL-1β decreased VEGF and MMP-3 expression, while treatment with IL-20 or BMP-2 antibodies decreased MCP-1, VEGF, and MMP-3 expression (Figure 3B-D). These data suggested that the upregulation of MMP-3 and VEGF was indeed mediated by IL-20, IL-1β, and BMP-2. “

I'm not 100% sure what the dots represent near the tops of the bars in the figures.

REPLY:

The dot plot as a representation of a distribution consists of group of data points plotted on a simple scale.

I would like to see the discussion section focusing on the discussion of the main findings. It begins by describing related concepts instead of the findings and what they may mean.

REPLY:

As suggested, we have deleted the describing related concepts. We also discussed and added these sentences “In the present study, we found many proinflammatory cytokines were upregulated in IVD cells under hypoxia conditions. IL-1β upregulated the expression of pro-inflammatory cytokines (IL-6 and IL-8), angiogenetic factor (VEGF), chemotactic factor (MCP-1), and disc degradation factor (MMP-3) in IVD cells under hypoxia conditions. IL-20 upregulated MCP-1 and VEGF expression. BMP-2 also upregulated the expression of MCP-1, VEGF, and IL-8 in IVD cells under hypoxia conditions. In addition, IL-1β modulated both the expression of IL-20 and BMP-2, but IL-20 only modulated BMP-2 either under a hypoxic or normoxic condition. Therefore, we concluded that the inflammation, chemotaxis, matrix degradation and angiogenesis after disc herniation are influenced by the hypoxic condition and controlled by IL-1β, IL-20, and BMP-2.” in the Discussion section of the revised manuscripts (Lines, 221-229).

I'd like to see the authors adding a section that acknowledges the limitations of the study

REPLY:

Thanks for your suggestion, and we have added a paragraph of description about the limitation in the revised manuscript as follows: (Lines, 280-288)

There were some limitations in this study: First, the intervertebral disc theoretically bears the axial force under gravity in the actual situation, so the IVD cells should be compressed by the axial load to mimic the loading of gravity, therefore, a future study is needed to investigate the cyclic compressive or tensile stress on the IVD cells. Second, we investigated the effect of IL-1β, IL-20, and BMP-2 on the expression of pro-inflammatory cytokines, chemotaxis factor, and angiogenesis factor of IVD cells under hypoxia. Our data is an accumulation of phenomenology; however, the weakness of this research was that we did not investigate the signaling pathway or possible molecular mechanism to address the roles of these cytokines in the pathogenesis of HIVD, which awaits future investigation.”

There are no references of material after 2014 which suggests that the material discussed may not be current. Are there no more recent studies that are relevant to this topic area?

REPLY:

As suggested, we added six more recent studies in the references of the revised manuscript (references: 6, 7, 8, 9, 31, and 34)

Round 2

Reviewer 1 Report

I was almost satisfied with your revised manuscript. Especially , you have stated your limitation sincerely in the manuscript.  

I would only suggest to revise Figure 2 and remove the concentration of each cytokine. The concentration of cytokine was described figure legends.

Author Response

Response to Reviewer 1 Comments

Point 1:

I was almost satisfied with your revised manuscript. Especially, you have stated your limitation sincerely in the manuscript. 

I would only suggest to revise Figure 2 and remove the concentration of each cytokine. The concentration of cytokine was described figure legends.

Response 1:

Thank you very much for your advice. My colleagues and I agree that the concentration of each cytokine in the figure 2 should be removed, and has been described in the figure legends according to your valuable suggestions. (Lines, 168-173)

Figure 2. Effects of IL-1β, IL-20, and BMP-2 in IVD cells under hypoxia. IVD cells were exposed to PBS, IL-1β (10 ng/ml), IL-20 (200 ng/ml), and BMP-2 (200 ng/ml) under hypoxia for 6 hours. (A-F) Total RNA was then isolated for RT-PCR with specific primers. β-actin was an internal control. *P < 0.05 compared to PBS-treated controls. Data are expressed as mean ± SD and are representative of three independent experiments.

Please see the attachment. Thank you very much.
